# Effect of Solvent Treatment on the Composition and Structure of Santanghu Long Flame Coal and Its Rapid Pyrolysis Products

**DOI:** 10.3390/molecules28207074

**Published:** 2023-10-13

**Authors:** Jia Guo, Meixia Zhu, Wenlong Mo, Yanxiong Wang, Junrong Yuan, Ronglan Wu, Junmin Niu, Kongjun Ma, Wencang Guo, Xianyong Wei, Xing Fan, Naeem Akram

**Affiliations:** 1Xinjiang Energy Co., Ltd., Urumqi 830000, China; gj459824057@126.com (J.G.); hjgwc@126.com (W.G.); 2State Key Laboratory of Chemistry and Utilization of Carbon Based Energy Resources and Key Laboratory of Coal Clean Conversion & Chemical Engineering Process (Xinjiang Uyghur Autonomous Region), School of Chemical Engineering and Technology, Xinjiang University, Urumqi 830046, China; 18580890109@163.com (M.Z.); 18599119007@163.com (Y.W.); yuanjunrong321@163.com (J.Y.); wuronglan@163.com (R.W.); kjma@xju.edu.cn (K.M.); wei_xianyong@163.com (X.W.); fanxing@sdust.edu.cn (X.F.); 3Key Laboratory of Coal Processing and Efficient Utilization, Ministry of Education, China University of Mining & Technology, Xuzhou 221116, China; 4School of Chemical Engineering, Minhaj University Lahore, Lahore 54000, Pakistan; naeemakram63@gmail.com

**Keywords:** solvent treatment, coal structure, products distribution, Py-GC/MS

## Abstract

Easily soluble organic components in Santanghu long flame coal (SLFC) from Hami (Xinjiang, China) were separated by CS_2_ and acetone mixed solvent (*v*/*v* = 1:1) under ultrasonic condition, and the extract residue was stratified by carbon tetrachloride to obtain the light raffinate component (SLFC-L). The effect of solvent treatment on the composition and structure of the coal and its rapid pyrolysis products was analyzed. Solvent treatment can reduce the moisture content in coal from 9.48% to 6.45% and increase the volatile matter from 26.59% to 28.78%, while the macromolecular structure of the coal changed slightly, demonstrating the stability of coal’s complex organic structure. Compared with raw coal, the relative contents of oxygen-containing functional groups and aromatic groups in SLFC-L are higher, and the weight loss rates of both SLFC and SLFC-L reached the maximum at about 450 °C. In contrast, the loss rate of SLFC-L is more obvious, being 33.62% higher than that of SLFC. Pyrolysis products from SLFC at 450 °C by Py-GC/MS are mainly aliphatic hydrocarbons and oxygenated compounds, and the relative contents of aliphatic hydrocarbons decreased from 48.48% to 36.13%, while the contents of oxygenates increased from 39.07% to 44.95%. Overall, the composition and functional group in the coal sample were changed after solvent treatment, resulting in a difference in the composition and distribution of its pyrolysis products.

## 1. Introduction

In recent years, coal has accounted for 70% of China’s disposable energy and 60% of China’s energy consumption, indicating that coal still plays a leading role in the national energy resources. In China, medium and low-rank coal (i.e., lignite and sub-bituminous coal) comprises nearly half of the national total coal reserves [1,2,3]. The Santanghu coalfield is one of the important coal production and processing bases in Hami, Xinjiang, China, with long flame coal (a low-rank coal) dominating, whose coalification degree is only higher than that of lignite.

Pyrolysis is one of the most important processing and utilization methods for coal and biomass, which is the basic step for combustion, gasification, carbonization, and liquefaction processes [4,5]. Pyrolysis technology can not only produce clean fuels but also generate valued coal-based products [6]. However, the industrial application of low-rank coal has been restricted due to its low calorific value, easy spontaneous combustion, and poor thermal stability [7,8,9,10]. Furthermore, due to the high volatile content of low-rank coal, more coal tar can be obtained by the pyrolysis process, which is of great significance for promoting the clean utilization of low-rank coal [11,12].

Solvent extraction is an effective technology to analyze the physical and chemical structure of coal, and this method makes it possible to change the use mode of low-rank coal from “high pollution, low efficiency, and low-added value” to “clean, high efficiency and high-added value”. Zou et al. [13] used tetrahydrofuran (THF) to extract organic components in two kinds of lignite and concluded that tetrahydrofuran extraction improved the pyrolysis activity of the coal. Zhu et al. [14] studied the influence of extractable compounds on the structure and pyrolysis behavior of Naomaohu lignite and Hutubi bituminous coal. It was found that solvent extraction with pyridine increased the pore diameter of the coal. Liu et al. [15] extracted Shengli lignite and Pingshuo bituminous coal with n-hexane and found that solvent treatment caused serious functional group evolution of coal. Zheng et al. [16] studied the influence of solvent treatment with hydrochloric acid, tetrahydrofuran, and carbon disulfide on the pore structure parameters and fractal characteristics of coal, and the results showed that solvent treatment improved the pore structure of coal. Mo et al. [17] used petroleum ether, methanol, carbon disulfide, and other solvents to perform five-stage extraction of Hefeng sub-bituminous coal. Results showed that methanol presented a higher extraction yield, while carbon disulfide/acetone (equal volume mixed solvent) was more conducive to the dissolution and diffusion of alcohol compounds. Wu et al. [18] found that the carbon disulfide/acetone mixed solvent can effectively extract and divide coal into soluble components and extract residue, and the method of density difference by carbon tetrachloride can be used to separate the inorganic minerals from the residue, and thus, light raffinate coal can be obtained. Hu et al. [19] discovered that sequential thermal dissolution (with cyclohexane, ethanol, and isopropanol as the solvent) could effectively remove heteroatoms, such as O, N, and S from Naomaohu lignite (NL), which may decrease NO_x_ and SO_x_ emission during coal pyrolysis, combustion, and gasification. The dissolution process is beneficial for the pyrolysis of NL and can break some non-covalent bonds and weaken some covalent bonds. Kan et al. [20] used mechanical activation, ultrasonic radiation, and microwave radiation enhancement methods with carbon disulfide/acetone as the solvent to swell Hefeng sub-bituminous coal. It was found that the internal pore structure of the coal was looser, resulting in lower mass transfer resistance and easier release of volatile components in the pyrolysis process.

Normally, owing to the wide variation in the structure and pyrolysis performance of different coal, there are still several limitations in understanding the reaction mechanism of the coal pyrolysis process. Besides, the relationship between the structure and pyrolysis reactivity has rarely been reported due to the complexity of coal’s structure. Qiang et al. [21] studied the effect of hydrothermal pretreatment, tetrahydrofuran pretreatment, and methane thermal pretreatment on the composition and structure of Shendong coal, and the physicochemical structure and pyrolysis properties of the pretreated coal samples were observed. Results showed that different treatment methods showed different effects on the change of functional group and pore structure, and these changes ultimately altered the composition of pyrolysis products. Therefore, elucidating the effect of solvent treatment on the composition and structural changes of coal, as well as the distribution of its pyrolysis products, is of great significance for the clean and efficient utilization of coal.

In order to study the influence of solvent treatment on the composition and structure characteristics of Santanghu long flame coal and its rapid pyrolysis products, the coal sample was firstly extracted with isometric CS_2_/acetone mixed solvent under mild conditions, and then light raffinate coal was obtained with carbon tetrachloride as the flotation agent, and the difference in the type and distribution of the pyrolysis products from the raw coal and its light raffinate will be analyzed in this paper.

## 2. Results and Discussion

### 2.1. Proximate and Ultimate Analyses

Proximate and ultimate analyses of SLFC and SLFC-L are shown in Table 1. The moisture (M_ad_) and ash (A_ad_) contents in SLFC-L were 6.45% and 3.92%, respectively, lower than that of SLFC, indicating that the solvent treatment process can dissolve some small molecules (such as H_2_O and low molecular compounds) and destroy some intermolecular forces in the coal. Meanwhile, the volatile matter (V_ad_) increased from 26.59% to 28.78% after the process of solvent treatment, which might be due to that the coal was swelled in the treated process, reducing the escape resistance of volatile matter and increasing its yield.

The contents of C, H, O, N, and S in SLFC and SLFC-L are shown in Table 1. After solvent treatment, the content of all elements, except O, decreased, which may be due to the fact that the lighter components in the coal mainly being removed during the treatment, while the heavier components, including some oxygen-containing compounds, were retained. On the other hand, the S content was obviously reduced, illustrating that the solvent treatment can remove some S-containing compounds.

As shown in Table 1, the H/C ratio of the swelled sample was almost unchanged, indicating that the main structure of organic matter in SLFC-L was slightly different from SLFC, and it also shows that IMCDSAMS can dissolve some compounds containing aromatic rings, and the dissolved compounds may be cyclic or unsaturated molecules.

Figure 1 presents the surface morphology of SLFC and SLFC-L. SLFC surface was smooth and dense, with some small particles attached, while the surface of SLFC-L was loose, and particle stacking can be observed. It could be inferred that solvent treatment made the coal structure looser, increasing the escape ability of low boiling point organic compounds.

### 2.2. FTIR Analysis

Figure 2 shows the FTIR spectra of SLFC and SLFC-L. According to some literature [22,23,24], the functional groups of coal samples can be divided into hydroxyl groups (3600–3000 cm^−1^), aliphatic groups (3000–2800 cm^−1^), oxygen-containing groups (1800–1000 cm^−1^), and aromatic ring groups (900–700 cm^−1^).

As exhibited in Figure 2, compared with SLFC, SLFC-L has different peak intensities at 3700–3200 cm^−1^, 2919 cm^−1^, 1694 cm^−1^, 1586 cm^−1^, 1227 cm^−1^, and 751 cm^−1^. The absorption peak observed near 3692 cm^−1^ is attributed to the non-associative free -OH-stretching vibration, and 3401 cm^−1^ is the absorption band of the intermolecular hydroxyl hydrogen bond.

The absorption peak of SLFC-L at 3401 cm^−1^ was enhanced, indicating that extraction treatment could increase the relative content of the hydroxyl functional group, which might be derived from the fact that the used solvent can break hydrogen bonds, such as OH-π, self-associated -OH, and OH-ether bonds [25]. Destruction of the hydrogen bonds could produce more independent hydroxyl groups, and the intensity of the absorption peak at 2919 cm^−1^ is also enhanced, indicating that SLFC-L contains more aliphatic structures. The absorption peak at 1694 cm^−1^ corresponds to the conjugated C=O vibration, the intensity of which is increased in SLFC-L, showing that the extraction treatment increases the oxygen-containing functional groups in the coal; 1650–1450 cm^−1^ was the skeleton vibration of benzene rings and heteroaromatic rings. The peak intensity of SLFC-L was higher at 1586 cm^−1^, which might be derived from the aromatic ring functional groups being retained in the coal after the extraction process, and the peak between 900–700 cm^−1^ is the characteristic absorption peak of substituted aromatic rings. It can be seen from the FTIR profiles that the peak intensity of SLFC-L at 751 cm^−1^ was significantly increased, demonstrating that SLFC-L contains more aromatic tri-substituted structures.

FTIR spectra of SLFC and SLFC-L, presented in Figure 2, were fitted by Peakfit 4 software, and the corresponding semi-quantitative analysis was performed. Results are displayed in Figure 3 and Table 2.

It should be noted that the relationship between increasing rate and area percentage in Table 2 is as follows:IR = [((SLFC-L)_ap_ − SLFC_ap_)/SLFC_ap_] × 100%
where IR and ap are the increasing rate and the area percentage, respectively.

After extraction–stratification treatment, compared with SLFC, the content of functional groups, such as self-associated -OH and OH-ether in SLFC-L, significantly decreased, resulting in some negative percentages in Table 2.

As shown in Table 2, compared with SLFC, the content (percentage of peak area) of self-associated -OH and OH-ether in SLFC-L decreased significantly, while the content of OH-π and cyclic -OH increased by 20.38% and 36.16%, respectively. It is speculated that solvent extraction destroys the molecular structure of coal and makes the self-associating -OH and OH-ether exist in the form of OH-π and cyclic -OH.

In the range of 3000–2800 cm^−1^, asymmetric aliphatic -CH_2_ dominates in SLFC and SLFC-L, accounting for more than 40%. As Table 2 summarized, the content of symmetric aliphatic -CH_2_ in SLFC-L was 31.47%, nearly the same as SLFC, while the content of aliphatic -CH and aliphatic -CH_3_ in SLFC-L was high, at 14% and 10%. The ratio of asymmetric stretching vibration peak areas of -CH_2_ and -CH_3_ in the sample can more reasonably characterize the relative content of each structure in organic matter. That is, the aliphatic structural parameter S = A(CH_2_)/A(CH_3_), and the parameter value of SLFC (7.56) > SLFC-L (4.45), indicating that aliphatic hydrocarbons in the former exist in the form of long chain, and the degree of side-chains is lower. In 1800–1000 cm^−1^, the content of conjugated C=O in SLFC-L increased significantly, indicating that the oxygen-containing compounds retained in SLFC-L may be increased after the solvent extraction process, and the content of aromatic C=C decreased by 55.44%, which may be due to the extraction of aromatic compounds in SLFC by carbon disulfide.

In addition, peaks around 900–700 cm^−1^ were mainly assigned to 3-substituted benzene rings, and the content of 5-substituted benzene ring increased significantly, while the content of 4-substituted benzene ring and 2-substituted benzene ring decreased; this indicates that, compared to SLFC, there are more side chains in the aromatic structural units of SLFC-L to be exposed.

### 2.3. TG-DTG Analysis

The weight loss behavior of the two samples was tested, and the profiles are shown in Figure 4.

As shown in Figure 4, the two samples presented similar mass loss trends from 30 °C to 1000 °C. From Figure 4a, it can be observed that at around 350 °C, the mass reduction of SLFC-L and SLFC began to intensify. The mass of SLFC-L was reduced to 40.18%, while the mass of SLFC was reduced to 30.07%. Compared with SLFC, SLFC-L presented a larger mass loss, which was due to the solvent treatment making the pore structure of the sample loose and expansive, increasing the escape ability of organic compounds in the removal process. This conclusion is consistent with the surface morphology of the coal sample in Figure 1.

From the DTG curve, as the temperature increased from 30 °C to 110 °C, mainly water and small molecules in the sample escaped. At 150–300 °C, decarboxylation may occur, or the high boiling point organics in SLFC and SLFC-L may be passed on, or small molecules adsorbed in the sample by capillary action may be removed. The temperature of the first peak is around 60 °C, indicating that the dehydration processes of the two coal samples mainly took place at this point. As illustrated in Figure 4b, SLFC-L presents an obvious weight loss rate peak at 215 °C, which may be due to the swollen and loose effect of the solvent treatment. The extraction–stratification process increases the internal pore diameter of coal, changes the mobility of small molecules in coal, reduces the mass transfer resistance, and improves the diffusion ability of organic matter in the sample [14,25].

It can also be seen from Figure 4b that the temperature of the maximum mass loss rate peak for SLFC and SLFC-L was also located around 450 °C, indicating that the pyrolysis of the two coal samples is more intense at this temperature, which is the main pyrolysis stage of the organic matter in coal. At temperatures over 600 °C, the polycondensation of the aromatic ring structure was dominant, and the concentrated small molecules (such as H_2_O and CO_2_) might be released.

### 2.4. Py-GC/MS Analysis

As shown in Figure 5 and Figure 6, 66 and 63 organic compounds can respectively be detected by Py-GC/MS from the pyrolysis of SLFC and SLFC-L at 450 °C, and these compounds can be classified into five groups, including alkanes, olefins, aromatics, oxygenated compounds (OCOCs) and compounds containing other heteroatoms (OHACOCs). The content distribution of the five groups is shown in Figure 7.

The main pyrolysis products from raw coal at 450 °C are aliphatic hydrocarbons and OCOCs. Among them, alkanes showed the highest content of 34.76% in aliphatic hydrocarbons. After extraction–stratification treatment, the content of aliphatic hydrocarbons in the pyrolysis products of SLFC-L was only 36.13%, lower than that of SLFC (48.48%). OCOCs increased from 39.07% to 44.95%, and OHACOCs also increased.

Appendix A and Figure 8 list all the detected compounds and give the corresponding classes. As demonstrated in Appendix A and Figure 8, alkanes from coal by pyrolysis were mainly divided into normal, branched, and cyclic alkanes, in which normal alkanes were dominant, with a significant decrease after the extraction process. The aromatics are mainly monocyclic and polycyclic aromatics. The content of monocyclic aromatics in the pyrolysis products after solvent treatment was significantly reduced, which may be due to the separation of some monocyclic aromatic compounds in the extraction process. It is evident from Figure 9 that the carbon number of alkanes in SLFC and SLFC-L ranged from 10 to 44, indicating that the extraction–stratification process is mainly a physical process and has not significantly damaged the macromolecular network structure of coal. The alkanes might have originated from the intrinsic components in coal or alkanoic acid decarboxylation from coal [19]. 

As illustrated in Appendix A, OCOCs mainly existed in the form of alcohols, aldehydes, carboxylic acids, and ethers, among which alcohols play a dominant role. There are eight kinds of alcohol compounds in the pyrolysis products of SLFC, with a relative content of 25.71%, while nine kinds of alcohols could be detected in the pyrolysis products of SLFC-L, with a content of 33.02%. Therefore, it can be speculated that oxygen-containing small molecular species in SLFC can be destroyed by IMCDSAMS and released during rapid pyrolysis.

## 3. Experimental

### 3.1. Raw Materials

Santanghu long flame coal (SLFC) was collected from the Santanghu coalfield in Hami, Xinjiang, China, and the coal was pulverized to pass through a 200-mesh sieve (<74 μm). The sample was dried naturally for 24 h and put in a drying dish for storage. Acetone, carbon disulfide, and carbon tetrachloride were purchased from Xilong Science Co., Ltd., Chengdu, Sichuan, China, Xin Bote Chemical Co., Ltd., Tianjin, China, and Hongyan Chemical Reagent Factory, Hedong District, Tianjin, China, respectively. All the solvents are analytical reagents and were distilled in a rotary evaporator before use.

### 3.2. Light Raffinate Component

The solvent treatment process of Santanghu long flame coal is shown in Figure 10. SLFC (15 g) was added to a 500 mL beaker with 300 mL isometric acetone and carbon disulfide mixed solvent (IMCDSAMS) into it, then the beaker was placed in ultrasonic equipment for 40 min and stood for 10 min. The extraction process was repeated several times until the filtrate was nearly colorless to ensure that the soluble components were extracted exhaustively. The mixture, including the solvent, the soluble portion from coal, and the insoluble part, was filtered to obtain filtrate and residue. The solvent was recycled from the filtrate by rotary evaporator.

As displayed in Figure 10, the insoluble part was stratified with 375 mL carbon tetrachloride under the condition of ultrasonic radiation for 0.5 h. The new liquid-solid mixture was put into a separatory funnel and followed by standing for 1 h. Light residue (LR) and heavy residue (HR) were clearly isolated, and they were released in sequence from the bottom of the funnel. As the LR was thoroughly extracted with IMACDSMS, it was filtered and dried to obtain the light raffinate component, which was labeled as SLFC-L.

### 3.3. Analysis Methods

In accordance with GB/T212-2008 [19] from China, the moisture, ash, volatile and fixed carbon in SLFC and SLFC-L were analyzed. Vario El III element equipment (Elemental UNICUBE, Frankfurt, Germany) was used to test the content of C, H, N, and S elements in the samples, and the content of O was calculated by difference [26]. Scanning electron microscopy (TESCAN MIRA4, Brno, Czech Republic) was carried out to characterize the surface morphology of SLFC and SLFC-L. An SDTQ-600 thermogravimetric analyzer (TA Instruments Limited, New Castle, DL, USA) was used to test the weight loss behavior of the samples, and the temperature was set from room temperature to 1000 °C with the heating rate of 10 °C/min with high-purity nitrogen as the carrier gas. Functional groups in the samples were analyzed by VERTEX 70 RAMI Fourier (Bruker Instruments GmbH, Ettlingen, Germany) infrared spectrometer. The wavelength range was selected as 400–4000 cm^−1^ with a resolution of 4 cm^−1^, and the wavenumber accuracy was 0.01 cm^−1^. The samples were dried for 10 h under vacuum conditions. A Sample and KBr mixture with a mass ratio of 1:160 was used to detect the functional groups in each experiment. Group components in the pyrolysis products from SLFC and SLFC-L were qualitatively and quantitatively analyzed at the temperature of 450 °C using pyrolysis–gas chromatography/mass spectrometry from the Elementar company in Tokyo, Japan. This instrument is composed of an EGA/PY-3030D thermal cracker (Frontier Corporation, Tokyo, Japan) and Agilent 7890/5795 GC/MS (Elemental Company in Tokyo, Japan). 0.5 mg of the coal sample was put in the pyrolysis tube and pyrolyzed at 450 °C for 12 s. The volatile products were analyzed online through GC/MS. The carrier gas was high-purity nitrogen (≥99.999%) with a flow rate of 1 mL/min and an inlet temperature of 200 °C. The heating program is maintained at 60 °C for 5 min and heated to 300 °C at a heating rate of 10 °C/min. The mass scan range was 30–500 *m*/*z*, and the split ratio was set to 20:1.

## 4. Conclusions

Santanghu long flame coal (SLFC) from Hami, Xinjiang (China) was treated by solvent extraction to obtain the corresponding light raffinate (SLFC-L). According to the TG-DTG profile, the weight loss rate of both SLFC and SLFC-L reached its maximum at the same temperature (450 °C), indicating the stability of the macrostructure network of coal. Py-GC/MS results showed that the pyrolysis products of SLFC and SLFC-L at 450 °C are mainly aliphatic hydrocarbons and oxygenated organic compounds. The hydrocarbons can be used as clean liquid fuels, and oxygenated organic compounds can be separated to obtain highly valued chemicals. In addition, the differences in the composition of pyrolysis products of SLFC and CLFC-L samples and degradation of the material of coal samples after extraction may prove significant changes in the macromolecular network structure of coal, which will be revealed in our future work.

## Figures and Tables

**Figure 1 molecules-28-07074-f001:**
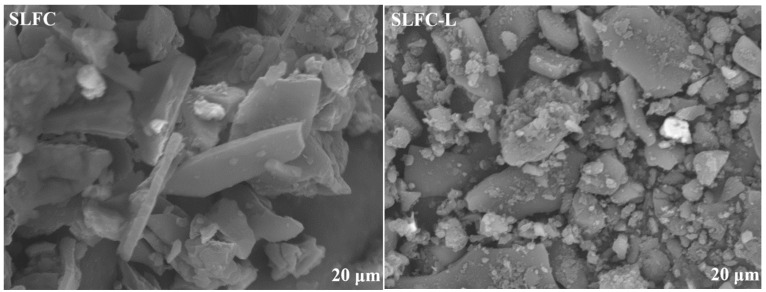
SEM morphologies of SLFC and SLFC-L.

**Figure 2 molecules-28-07074-f002:**
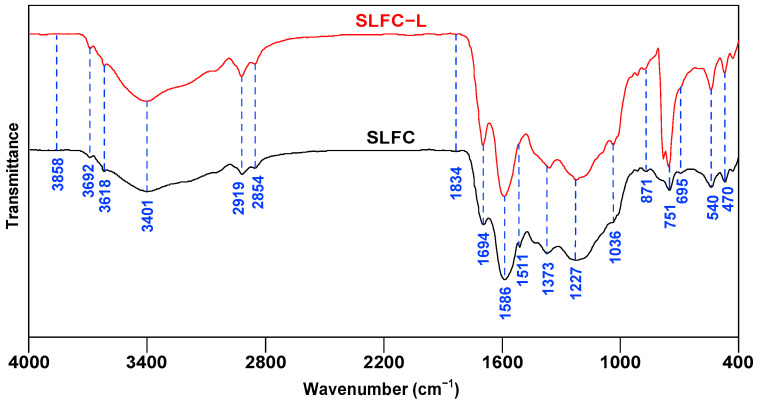
FTIR spectra of SLFC and SLFC-L.

**Figure 3 molecules-28-07074-f003:**
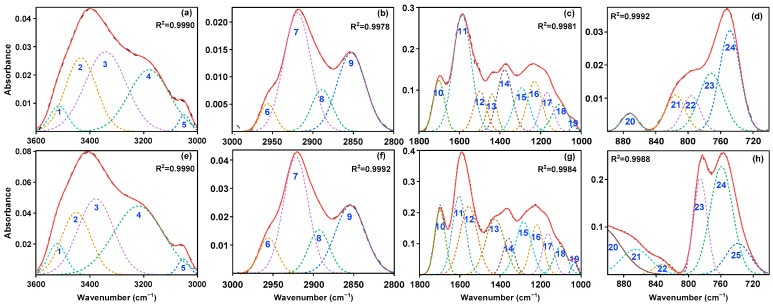
FTIR fitting results of SLFC (**a**–**d**) and SLFC-L (**e**–**h**). Solid line: raw data; Dashed line: Fit data.

**Figure 4 molecules-28-07074-f004:**
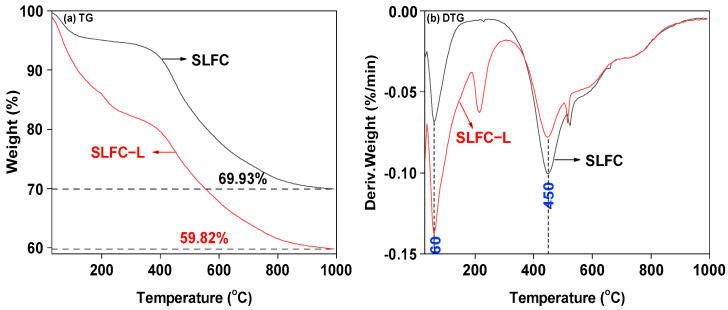
TG-DTG profiles of SLFC and SLFC-L.

**Figure 5 molecules-28-07074-f005:**
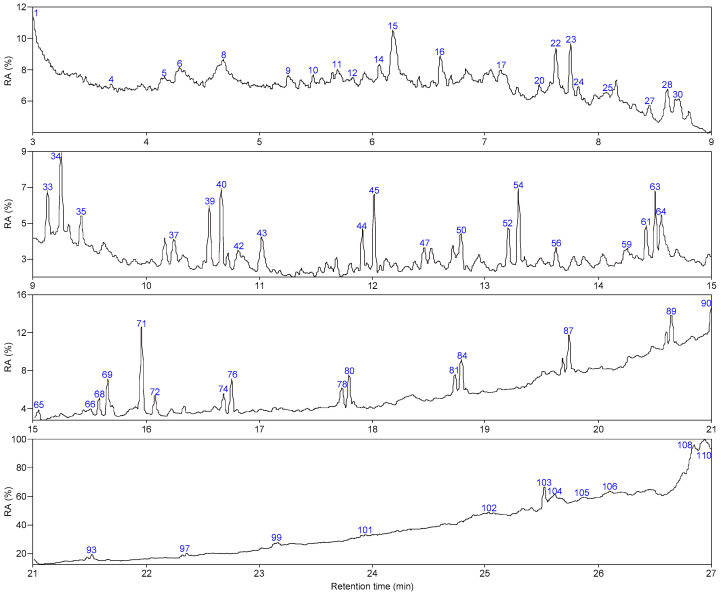
Total ion chromatogram of SLFC.

**Figure 6 molecules-28-07074-f006:**
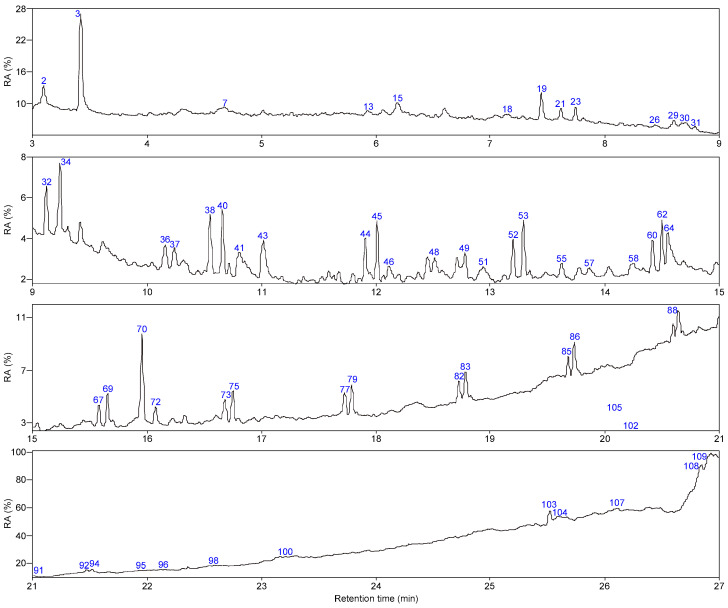
Total ion chromatogram of SLFC-L.

**Figure 7 molecules-28-07074-f007:**
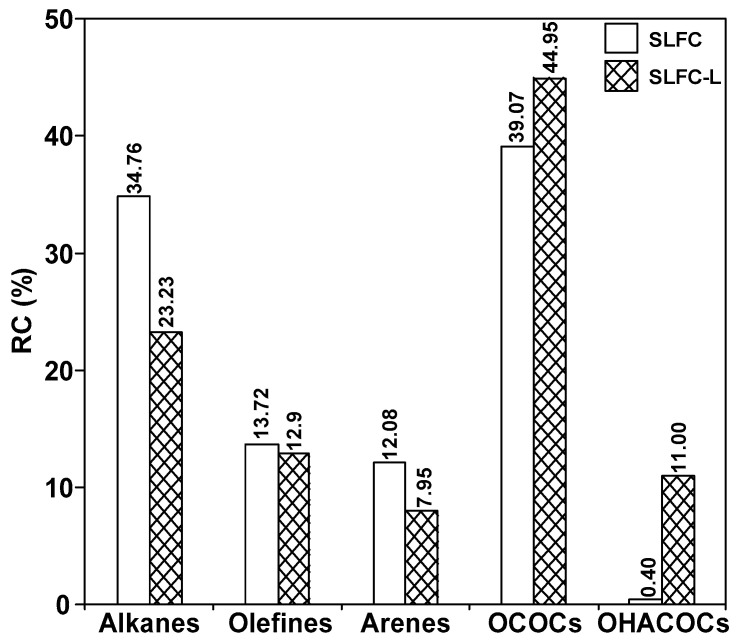
The relative content of each component in SLFC and SLFC-L.

**Figure 8 molecules-28-07074-f008:**
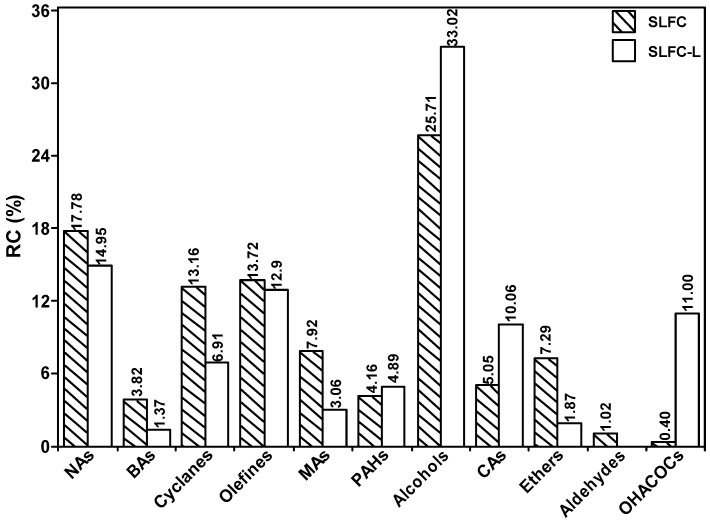
Distribution of group components in pyrolysis products from SLFC and SLFC-L.

**Figure 9 molecules-28-07074-f009:**
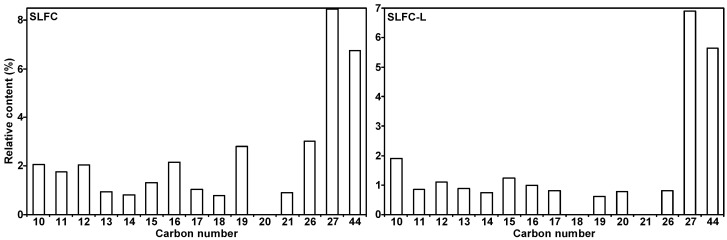
Distribution of alkanes in the resulting volatiles from SLFC and SLFC-L.

**Figure 10 molecules-28-07074-f010:**
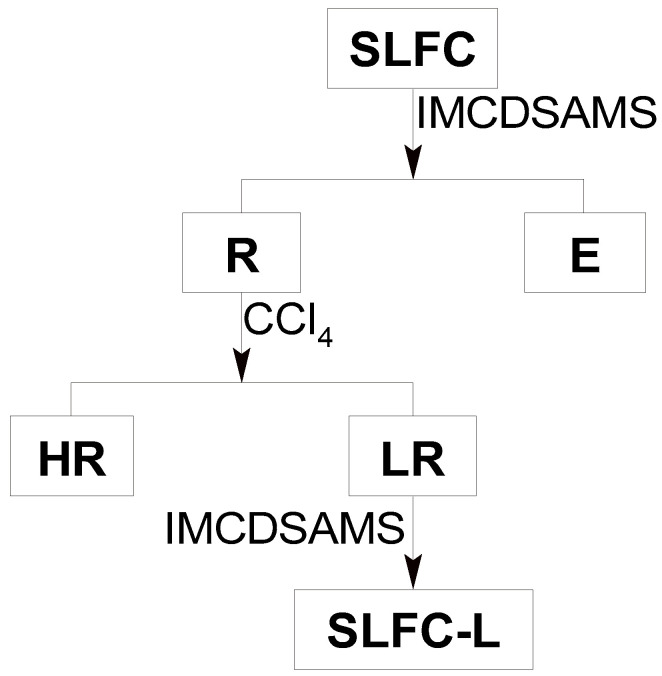
Procedure for extraction–stratification of SLFC.

**Table 1 molecules-28-07074-t001:** Proximate and ultimate analyses of the coal samples.

Sample	Proximate Analysis (wt.%)				Ultimate Analysis (daf, wt.%)					H/C
	M_ad_	A_ad_	V_ad_	FC_ad_	C	H	N	S	O^diff^	
SLFC	9.48	5.81	26.59	58.12	69.10	3.39	0.91	0.72	25.88	0.59
SLFC-L	6.45	3.92	28.78	60.85	63.24	3.18	0.85	0.60	32.13	0.60

_ad_: air dry basis; daf: dry and ash-free basis; ^diff^: by difference.

**Table 2 molecules-28-07074-t002:** Relative content of functional groups for the samples.

Number	Band Position/cm^−1^	Functional Group	Area Percentage/%		Increasing Rate/%
			SLFC	SLFC-L	
1	3600–3500	OH-π	5.30	6.38	20.38
2	3500–3350	Self-associated OH	26.11	20.94	−19.80
3	3350–3260	OH-ether	37.50	31.19	−16.83
4	3260–3170	Cyclic OH	28.93	39.39	36.16
5	3170–3000	OH-N	2.17	2.09	−3.69
6	3000–2930	Aliphatic -CH_3_	6.28	10.00	59.24
7	2930–2900	Asymmetric aliphatic -CH_2_	47.46	44.53	−6.17
8	2900–2870	Aliphatic -CH	13.03	14.00	7.44
9	2870–2800	Symmetric aliphatic -CH_2_	33.23	31.47	−5.30
10	1800–1700	Carboxylic C=O	8.42	10.12	20.19
11	1700–1600	Conjugated C=O	-	15.60	-
12	1600–1480	Aromatic C=C	39.32	17.52	−55.44
13	1480–1400	Asymmetric CH_3_-, CH_2_-	5.51	15.58	182.76
14	1400–1240	Symmetric deformation -CH_3_	24.02	18.41	−23.36
15	1240–1160	C-O phenols	18.32	16.22	−11.46
16	1160–1090	Grease C-O	6.56	5.52	−15.85
17	1090–1030	Alkyl ethers	0.51	1.03	101.96
18	900–860	Five adjacent H deformation	5.56	36.05	548.38
19	860–810	Four adjacent H deformations	13.64	2.31	−83.06
20	810–750	Three adjacent H deformations	40.07	51.14	27.63
21	750–720	Two adjacent H deformations	40.72	10.50	−74.21

## Data Availability

Not applicable.

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
