# Peer review of "Effect of Solvent Treatment on the Composition and Structure of Santanghu Long Flame Coal and Its Rapid Pyrolysis Products"

_molecules, 2023, doi:10.3390/molecules28207074_

Round 1
Reviewer 1 Report
In this paper, the light raffinate component (SLFC-L) of Santanghu long flame coal (SLFC) was obtained by solvent extraction-stratification treatment, and the effects of solvent treatment on the composition, structure and rapid pyrolysis products distribution of SLFC and SLFC-L were investigated. This work presents importance in processing/cleaning of coal. The composition, structure and rapid pyrolysis products distribution of the coal and its light raffinate component were characterized in detail, and the methodology is well-described. The manuscript can be accepted after minor revision. However, the following problems need to be clarified.
1. In Figure 1, the expression of “Ultrasonic assisted extraction-stratification process of SLFC” should be modified as “Procedure for extraction-stratification of SLFC”.
2. In Table 1, replacing ‘elemental analysis’ and ‘a’ with ‘ultimate analysis’ and ‘diff’ may be more reasonable.
3. In Table 2, please explain why some percentages are negative? This is an interesting observation that readers want to understand.
4. The authors are also advised to highlight the novelty of the work since there have been many works reported in this area.
5. In line 161, the reason for the H/C value of the swelling sample changed slightly can be added with “It also shows that IMCDSAMS can dissolve some compounds containing aromatic rings, and the dissolved compounds may be cyclic or unsaturated molecules”.
Acceptable
Author Response
Thank you very much for taking the time to review this manuscript. Please find a detailed response below.

Reviewer 2 Report
The manuscript under review is aimed at a contrastive analysis of thermal behaviour of SLFC coal and its residue after extraction (SLFC-L). The changes in composition and structure had to be taken into account in the research.
Comments:
1. The manuscript under review does not present structural parameters data but focuses only on structural-chemical parameters on the basis of FT-IR data.
2. The manuscript lacks in results of determination of the mass loss of SLFC sample caused by extraction.
3. In Fig. 2 , the morphology of samples and changes in diameter of pores are not visible. Only visible there is grinding of material caused by extraction.
4. Fig. 4. In my opinion, the deconvolution of bands in FT-IR spectra was carried inconsistently.
In the ranges listed below, the lacks of subpeaks should be completed:
3600-3000 cm-1: the lack of v(C-H)ar near 3030 cm-1 [see Fuel 66(1987)973];
3600-3000 cm-1: the lack of subpeak from carboxylic acid dimmers [Fuproc 85(2004)815];
3000-2800 cm-1: as a rule, 5 subpeaks are distinguished [Fuel 63(1984)245];
1890-1505 cm-1: 7 subpeaks are distinguished [Fuel 77(1998)563];
During curve-fitting of band in FT-IR spectra, it would be better to present deconvolution of the entire fragment of spectrum in the range of 3680-2400 cm-1 [Fuel 96(2012)298].
5. Conclusions, lines 287-289 „The treatment method is mainly a physical process and does not significantly damage the macromolecular network structure of coal.” The manuscript lacks in experimental data about the parameters of macromolecular network structure of coal However, the differences in composition of pyrolysis products of SLFC and CLFC-L samples and degradation of the material of coal samples after extraction presented in Tables S1-S5 prove the significant damage of the macromolecular network structure of coal.
Author Response

(The authors gave the same response as above.)

Round 2
Reviewer 2 Report
The manuscript can be published in its present form